# CuS@Corn Stalk/Chitin Composite Hydrogel for Photodegradation and Antibacterial

**DOI:** 10.3390/polym11091393

**Published:** 2019-08-24

**Authors:** Yutong Xiong, Bichong Luo, Guixin Chen, Jihai Cai, Qimeng Jiang, Bin Gu, Xiaoying Wang

**Affiliations:** State Key Laboratory of Pulp and Paper Engineering, South China University of Technology, 381 Wushan Road, Guangzhou 510640, China

**Keywords:** CuS, corn stalk, hydrogel, photodegradation, antibacterial

## Abstract

Copper sulfide nanoparticles (CuS NPs) have recently attracted extensive attention in various fields due to their excellent optical and electrical properties. However, CuS NPs are easy to agglomerate in their preparation on account of the high surface activity. In this study, uniform dispersion of CuS NPs were fabricated with corn stalk as a template and stabilizer, further CuS@corn stalk/chitin composite hydrogel was obtained by crosslinking with chitin. The results reveal that the CuS NPs were evenly dispersed into the composite hydrogels with a three-dimensional network structure, which were verified by the UV-vis spectrum, XRD, FT-IR spectra and SEM. In addition, the as-prepared composite hydrogel with the traits of peroxidase-like activity can convert H_2_O_2_ into an extremely oxidative and toxic ·OH, which manifested good effects for photodegradation of RhB and antibacterial against *Escherichia coli* and *Staphylococcus aureus*. Hence, the composite hydrogels could be used for photocatalytic treatment and sterilization of wastewater, which provides a new idea for the functional application of CuS NPs.

## 1. Introduction

In recent decades, nanoscale materials have received considerable research interest due to their significant impact on many aspects of modern society [1]. Therein, nanoscale particles exhibit outstanding structural performance like excellent photoelectric properties compared with individual molecules or bulk form [2]. As a well-known p-type semiconductor with a band gap of 1.2–2.0 eV [3], CuS nanoparticles (CuS NPs) have been intensively studied in various fields due to their outstanding photoelectric properties, photo-thermal properties, and photocatalytic activity [4] than that of copper sulfide itself. Such excellent properties make it an ideal candidate for diverse applications such as photocatalytic degradation of organic pollutants [1], photo-thermal treatment of tumors [2], drug delivery [5], and tissue imaging [6]. It is reported that CuS NPs as semiconductor materials can generate electrons and holes with light excitation, inducing reactive oxygen species (ROS), which are proven to be effective organic oxides [7]. Thus, in the presence of H_2_O_2_, CuS NPs can catalyze the formation of ·OH, which leads to the oxidation of cell components, therefore inducing programmed cell death and inhibiting the growth of bacteria [8]. Similarly, they can degrade hazardous substances such as organic dyes in industrial production. However, due to their high surface activity, CuS NPs are easy to agglomerate together in the absence of a carrier and stabilizer, thus hindering their practical applications [9].

To avoid possible agglomeration and make full use of CuS NPs, surfactant is commonly involved to avoid the agglomeration of nanoparticles. This not only complicates the preparation methods, but also adds environmental load to society [10]. Hence, it is an ideal choice to develop a low-cost, clean, and non-toxic method to synthesize CuS NPs and expand its application field [11].

Recently, biomass is an abundant renewable bioresource with many advantages of biocompatibility and biodegradability. As biomacromolecules, the polyhydric structure of biomass form molecular level capsules through inter- and intramolecular hydrogen bonds, which offer a large number of active sites serving as not only a template for the growth of nanoparticles but also as a stabilizer to protect nanoparticles [10,12]. Our previous study used xylan [11], quaternized chitosan [13], lignin, and starch [9] as templates and stabilizers in preparation of CuS NPs. However, the separation and purification of the above-mentioned biomass can bring trouble to the quantitative production, causing potential environmental pollution [14]. Hence, direct utilization of biomass is a good idea compared with the complex separation and purification process.

Herein, as illustrated in Figure 1a, we employed a simple and facile method to prepare CuS NPs with corn stalk as a template and stabilizer; CuS@corn stalk/chitin composite hydrogel was further fabricated by crosslinking with chitin. The obtained composite hydrogel was applied for photodegradation of RhB and antibacterial against *Escherichia coli* and *Staphylococcus aureus*, for its peroxidase-like activity which could convert H_2_O_2_ into an extremely oxidable and toxic ·OH (Figure 1b).

## 2. Materials and Methods

### 2.1. Materials

Chitin was purchased from Aladdin Biochemical Technology Co., Ltd. (Shanghai, China) and treated with ball-milling for 8 h. Corn stalk was collected from agriculture waste in Jinan, China, and crushed to 40–60 mesh with a crusher, and then continuously milled for 10 h at 220 rpm with a planetary ball mill. All the chemicals and solvents used in the experiments were of analytical grade and used without further purification. Distilled water was used for preparation of all solutions.

### 2.2. Preparation of CuS@Corn Stalk/Chitin Composite Hydrogel

NaOH/urea (11%/4% wt) aqueous solution was used as a solvent for chitin and Corn Stalk. The chitin and corn stalk powder were dispersed into a NaOH/urea aqueous solution to obtain 6% wt chitin suspensions and 3% wt corn stalk suspensions, respectively. Subsequently, the suspensions were frozen below –40 °C overnight, and then thawed at 0 °C with stirring to obtain a chitin and corn stalk solution, respectively.

CuSO_4_·5H_2_O (0.152 g) was dissolved in distilled water (1.5 mL). Then, the diluted ammonia (7 M, 1 mL) was added to the CuSO4·5H2O solution under constant stirring. Deep blue copper-amine complex ([Cu (NH_3_)_4_]^2+^) was obtained after the precipitation of light blue basic copper sulfate disappeared [15]. After that, the [Cu (NH_3_)_4_]^2+^ solution was added to the above corn stalk/NaOH/Urea solution (12.5 g) and stirred for 0.5 h. Then, Na_2_S·9H_2_O (0.08 M, 2.5 mL) was added dropwise to this solution at room temperature and stirred for 20 min. Treated with ice water bathing, chitin alkali urea solution was added to the reaction system with stirring to make chitin disperse evenly. Epichlorohydrin (3 mL) was added at last as a crosslinking agent.

The above viscous mixed solution was poured into a 12-well plate, which was maintained at 4 °C for 24 h for the chemical cross-linking reaction of the hydroxyl groups on the chitin and stalk chains with epichlorohydrin (ECH). Finally, the obtained hydrogels were removed from the mold and immersed into the distilled water until the pH was neutral.

### 2.3. Characterization of CuS@corn Stalk/Chitin Composite Hydrogel

An EVO-18 scanning electron microscope (SEM, Zeiss, Oberkochen, Germany) was used to investigate the morphological and structural characteristics of the hydrogel. The FT-IR spectra were recorded on a Tensor 27 (Bruker, Karlsruhe, Germany) via a KBr pellet method under dry air at room temperature; each sample was scanned from 4000 to 500 cm^−1^ with a resolution of 4 cm^−1^. X-ray diffraction (XRD) analysis of the composite hydrogel crystal structure was performed in a D8 Advance X-ray diffractometer (Bruker, Germany) using Cu Kα radiation (λ = 0.15418 nm) at 40 kV with a scanning rate of 2°/min and a scanning scope of 20−80° (2θ). UV-Vis absorption spectra were obtained for determination of optical properties in a range of 200–800 nm by a Shimadzu UV-1800 spectrophotometer. The hydrogels used for the characterizations had been lyophilized for three days at –80 °C.

### 2.4. Peroxidase-Like Activity of CuS@Corn Stalk/Chitin Composite Hydrogel

The peroxidase-like activity experiment was performed at room temperature and the reaction time is 35 min. The study was carried out in four groups with the solution uniformly dispersed in the pH = 5.2 NaAc/Ac buffer (0.1 M): TMB, TMB+H_2_O_2_, TMB+H_2_O_2_+blank hydrogel, TMB + H_2_O_2_ + CuS@corn stalk/chitin composite hydrogel. The weight of hydrogels were 0.61 g and the final working concentrations were 832 μM and 0.1 M for TMB and H_2_O_2_, respectively. The UV-vis spectra were tested at 652 nm with a scan range of 800–350 nm.

### 2.5. Photocatalytic Activity of CuS@Corn Stalk/Chitin Composite Hydrogel

The hydrogel was immersed in a rhodamine B (RhB) solution (0.1 mM, 50 mL), followed by dropping 2 mL of H_2_O_2_ and then NIR laser (808 nm, 2 W/cm2) irradiated on the hydrogel through the RhB solution in the dark. The Optocouplers laser (MW-GX-808/1−5000 mW) was purchased from Leishi Optoelectronics Technology Co., Ltd. (Changchun, China). The RhB solution (3 mL) was removed out at given time interval (15 min) to test the UV−vis spectra by Shimadzu UV-1800 spectrophotometer with a scan range of 800–400 nm. The changes in the intensity of the absorbance at 552 nm were measured.

### 2.6. Antibacterial Experiments

The antibacterial property of the hydrogel was conducted in NaAc/Ac buffer (0.1M). *E. coli* was used as a representative Gram-negative bacterium and *S. aureus* represented for Gram-positive bacteria. *E. coli* and *S. aureus* were divided into four groups: (I) bacteria, (II) bacteria + H_2_O_2_, (III) bacteria + blank hydrogel + H_2_O_2_ and, (IV) bacteria + CuS@corn stalk/chitin composite hydrogel. The final working concentration of H_2_O_2_ was 0.1 M. The mixtures for all groups were reacted for 30 min, then placed on Luria-Bertani solid medium and incubated at 37 °C for 24 h. The size of the bacteriostatic zone was observed to assess the antibacterial effect of composite hydrogel.

## 3. Results and Discussion

### 3.1. Structure and Morphology of CuS@Corn Stalk/Chitin Composite Hydrogel

The obtained CuS@corn stalk was carried out by UV-vis absorption spectrum as is displayed in Figure 2a. It reveals the absorption edge to be at about 611 nm, which suggests that the sample is nanoscale in size. The spectrum also shows an increased absorption in the near-IR region, which is the characteristic of copper sulfide [16].

Figure 2b shows the FT-IR spectra of the composite hydrogel, corn stalk and chitin. In the FT-IR spectrum of chitin, the absorption peaks at 3442, 3105 cm^−1^ assigned to the stretching of –OH groups, –NH groups. The absorption peaks at 1658 cm^−1^, 1558 cm^−1^ and 1315 cm^−1^ belong to the amide I band (ν C=O), amide II band (ν N–H), and amide III band (ν C–N), respectively [17].

For corn stalk, the absorption peaks of –OH, C–H and C–O are similar to chitin. The stretching vibration absorption peaks of C=O appeared at 1733 cm^−1^, and the stretching vibration of C–H stretching out of plane of aromatic ring appeared at 831 cm^−1^ [18]. After the synthesis of CuS, CuS@corn stalk showed weaker peaks than pure corn stalk and C=O stretch vibration absorption disappear, indicating NaOH/urea aqueous solvent affects the structure of corn stalk to some extent.

It can be seen that most of the characteristic peaks of corn stalk and chitin have appeared in the spectrum of the composite hydrogel, but most of the peaks became weaker for the decrease of –OH and –NH groups among corn stalk and chitin, indicating the crosslinking between corn stalk, chitin and epichlorohydrin [19].

The scanning electron microscope was applied to investigate the morphology of the hydrogel. Figure 3a–d show the SEM images of the surface of the hydrogel. The three-dimensional network structures can be observed in Figure 3c, due to the crosslinking of corn stalk, chitin and epichlorohydrin as confirmed in FT-IR results. As seen in Figure 3a, in the absence of CuS NPs, the blank hydrogels revealed their porous structures. In comparison, the composite hydrogel in Figure 3b displayed a neat arrangement of fibrous structures while single fibers can be observed in 3d, which are attributed to CuS NPs inducing the formation of 3D network structures, and also indicate that the CuS NPs were uniformly distributed in the hydrogel. The presence of the CuS can be further confirmed through the EDS profile in Figure 3e, where certain amounts of copper and sulfur were detected. A whole piece of blank and composite hydrogels is shown in Figure 3f. The composite hydrogel has a dark green appearance which can help prove the existence of copper sulfide [20].

### 3.2. XRD Patterns of CuS@Corn Stalk/Chitin Composite Hydrogel

The XRD patterns of the CuS@corn stalk/chitin composite hydrogel are shown in Figure 4. Compared with the XRD pattern of the blank hydrogel, the characteristic diffraction peak of copper sulfide appeared in the pattern of the composite hydrogel, corresponding to (101) and (110) crystal faces, matching with the peaks in JCPDS (no. 06-0464). This indicates that CuS NPs were successfully deposited on corn stalk. Due to the natural three-dimensional network structures in hydrogel, it has a good coating effect on CuS NPs, thus the diffraction peak of copper sulfide is weak.

### 3.3. Peroxidase-Like Activity of CuS@Corn Stalk/Chitin Composite Hydrogel

According to reports, some inorganic nanomaterials such as Fe_3_O_4_ nanoparticles can simulate the properties of natural peroxidase and convert hydrogen peroxide into hydroxyl radicals which have strong oxidability [21]. Through experiments, we prove that CuS NPs in hydrogel have the same capability to induce hydroxyl radical formation.

As shown in Figure 5, the solution treated with H_2_O_2_ and CuS@corn stalk/chitin composite hydrogel exhibited well absorbance at 652nm (Figure 5a) and the solution turned into blue (Figure 5b IV), while the other groups did not, which confirms the formation of oxidation product 3,3′,5,5′-tetramethylbenzidine diimine (TMBDI) [22]. The oxidation reaction of 3,3′,5,5′-tetramethylbenzidine (TMB) would not happen when there was only H_2_O_2_ or blank hydrogel in the system, which demonstrates that CuS NPs in the hydrogel have the peroxidase-like activity as peroxidase mimics and can function as a catalyst to induce H_2_O_2_ produce ·OH.

### 3.4. Antibacterial Study of CuS@Corn Stalk/Chitin Composite Hydrogel

Figure 6 shows the antibacterial results of hydrogels against *E. coli* and *S. aureus* which represent for Gram-negative bacterium and Gram-positive bacterium respectively. It can be seen from Figure 6a (II), (III) and Figure 6b (II), (III) that both H_2_O_2_ with the given concentration and blank hydrogel without copper sulfide have no antibacterial property. The antibacterial effect of hydrogels was clearly shown in Figure 6a (IV) and Figure 6b (IV) with 27.53 ± 0.37 mm and 29.23 ± 0.67 mm bacteriostatic zone size respectively.

The above results can illustrate well that it is the copper sulfide in the hydrogel that acts as an antibacterial agent via stimulating H_2_O_2_ to release ·OH, which oxidized the cell components inducing the death of bacteria.

### 3.5. Photocatalytic Activity of CuS@Corn Stalk/Chitin Composite Hydrogel

Under the condition of near-infrared irradiation, CuS NPs induced hydrogen peroxide to produce hydroxyl radical, which acted as an oxidant to oxidize RhB and realized photodegradation. We evaluated the photocatalytic activity of hydrogels by detecting characteristic peaks at 553 nm.

It can be seen from Figure 7a that the prepared hydrogels have a good photocatalytic ability to degrade RhB, and in Figure 7b, the degradation of RhB by hydrogels shows a linear relationship with time, which verifies the stability of hydrogels in degrading organic dyes.

## 4. Conclusions

In this study, CuS NPs with good dispersion were synthesized by using corn stalk as the growth template and stabilizer, and then prepared CuS@corn stalk was added into chitin to obtain the composite hydrogel by epichlorohydrin crosslinking. In this way, corn stalk was directly utilized without further separation and purification, which reduced the consumption of energy and chemical agents. Furthermore, under the irradiation of light, the CuS NPs embedded in the hydrogel were catalytically active sites in the decomposition of hydrogen peroxide to generate the strong oxidant ·OH. Hence, the hydrogel exhibited good antibacterial properties against both *E. coli* and *S. aureus*, showing promising application in medicine and food preservation. Furthermore, the hydrogel catalyzed the degradation of organic dyes in an effective way, which provides a new idea for wastewater treatment in industry.

## Figures and Tables

**Figure 1 polymers-11-01393-f001:**
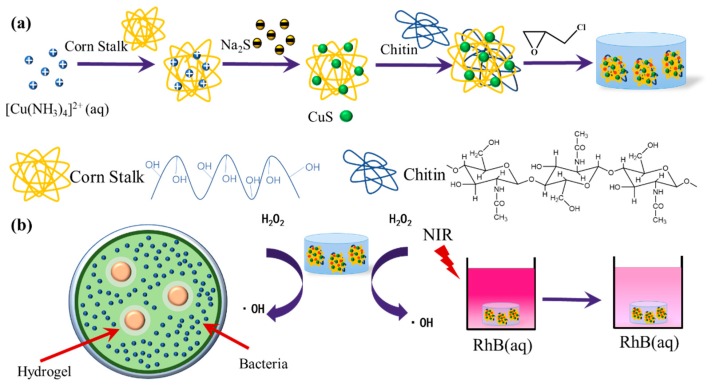
(**a**) Schematic diagram of the synthesis of CuS@corn stalk/chitin composite hydrogels; (**b**) the composite hydrogels used for antibacterial and photodegradation experiment.

**Figure 2 polymers-11-01393-f002:**
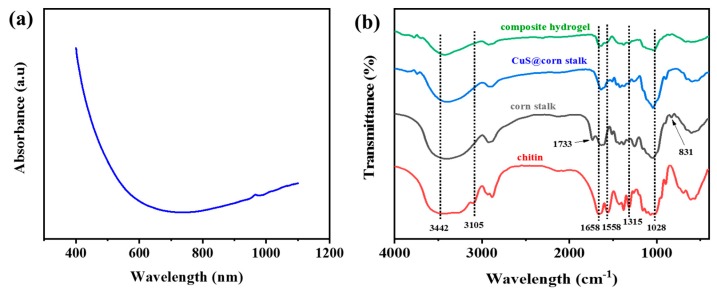
(**a**) UV-vis absorption spectrum of CuS; (**b**) FT-IR spectra of the composite hydrogel, CuS@corn stalk, corn stalk and chitin.

**Figure 3 polymers-11-01393-f003:**
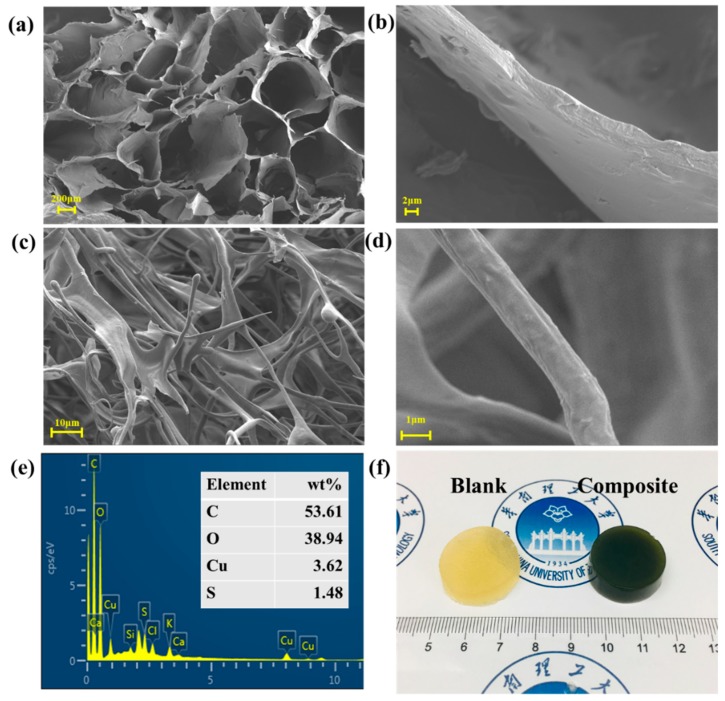
(**a**,**b**) SEM images of blank hydrogels; (**c**,**d**) SEM images of CuS@corn stalk/chitin composite hydrogels; (**e**) energy dispersive spectrometer (EDS) about the surface of CuS@corn stalk/chitin composite hydrogel; (**f**) digital photograph of blank hydrogel and composite hydrogel.

**Figure 4 polymers-11-01393-f004:**
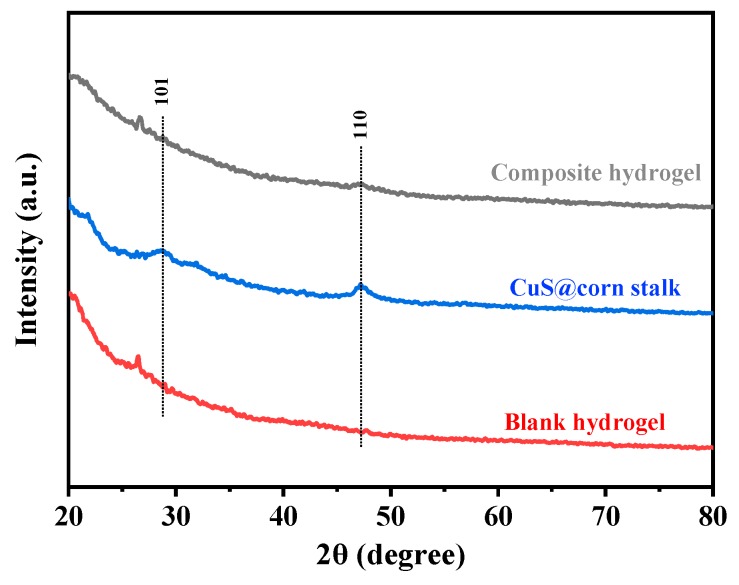
XRD patterns of CuS@corn stalk/chitin composite hydrogel, CuS@corn stalk and blank hydrogel.

**Figure 5 polymers-11-01393-f005:**
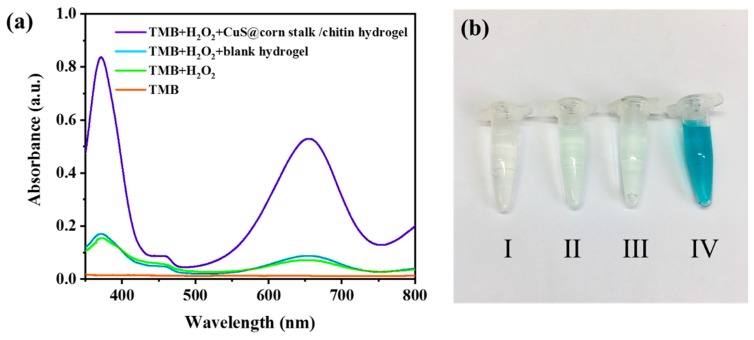
(**a**) UV/vis absorption spectra of different TMB solutions; (**b**) different TMB solutions: (I) TMB, (II) TMB + H_2_O_2_, (III) TMB + H_2_O_2_ + blank hydrogel, (IV) TMB + H_2_O_2_ + CuS@corn stalk/chitin composite hydrogel.

**Figure 6 polymers-11-01393-f006:**
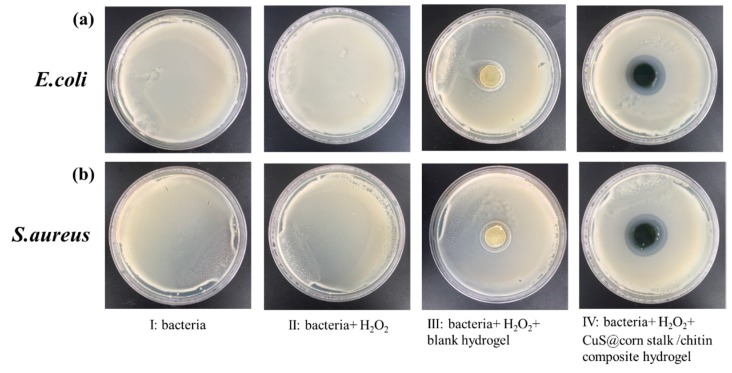
Photos of culture results of different bacterial solutions. (**a**) and (**b**) represent *E. coli* and *S. aureus*, respectively. I: Bacteria; II: Bacteria + H_2_O_2_; III: Bacteria + H_2_O_2_ + blank hydrogel; IV: Bacteria + H_2_O_2_ + CuS@corn stalk/chitin composite hydrogel.

**Figure 7 polymers-11-01393-f007:**
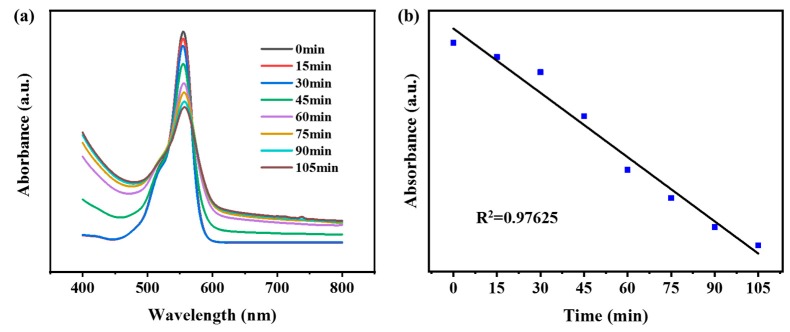
(**a**) UV/vis absorption spectra of RhB dye photodegradation in NIR laser with given time interval; (**b**) the linear fitting diagram of the peak at the wavelength of 553 nm.

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
