# Peer review of "CuS@Corn Stalk/Chitin Composite Hydrogel for Photodegradation and Antibacterial"

_polymers, 2019, doi:10.3390/polym11091393_

Round 1
Reviewer 1 Report
The manuscript “CuS@corn stalk/chitin Composite Hydrogel for 3 Photodegradation and Antibacterial” reported the fabrication of a uniform dispersion of CuS NPs with corn stalk as template and stabilizer, further CuS@corn stalk/chitin composite hydrogel was obtained by crosslinking with chitin. The composite hydrogel manifested good effects for photodegradation of RhB and antibacterial against E. coli and S. aureus.
The work was well-done, the manuscript well-organized, and the references are actualized.
I have only one doubt, why the authors did not study the effect of NIR in the antibacterial properties, since they have already proved the successful of the CuS nanoparticles under NIR irradiation in their former paper.
Moreover, some language correction would be appreciated.
Author Response
Reviewer #1: The work was well-done, the manuscript well-organized, and the references are actualized. I have only one doubt, why the authors did not study the effect of NIR in the antibacterial properties, since they have already proved the successful of the CuS nanoparticles under NIR irradiation in their former paper.
Author reply: Thank you for your good comments. CuS NPs in the composite hydrogel to generate ·OH is attributed to its photodynamic effect, that is, under NIR irradiation CuS NPs can catalyze H2O2 to ·OH, which acts as a strong oxidant to degrade RhB and against bacteria. But during the antibacterial experiment, we culture bacterial for 24h at ambient light. As is well-known that ambient light also contains infrared light, which is enough to excite CuS NPs to generate ·OH (called as peroxidase-like activity) against bacteria. Your proposal is very good and inspired me. In the following research, we will further combine the effect of NIR-induced heat and hydroxyl radical generation of Cu SNPs for synergistic antibacterial.
Reviewer 2 Report
The authors report a synthesis method for well dispersed CuS nanoparticles, using corn stalk as the growth template and stabilizer, and preparation of CuS/ corn stalk/ chitin hydrogel, using epichlorohydrin as crosslinker. The prepared hydrogel can be used for photocatalytic degradation of organic dyes and also for its antibacterial effect.
Some revisions are necessary:
At lines 41, 195, 208, 231: I suggest to the authors to use the same sign for OH radical (e.g. •) in the whole manuscript. At lines 64, 126, 201: I suggest to the authors to use italic font for all bacteria names (e.g. Escherichia coli and Staphylococcus aureus) in the whole manuscript. At page 2, line 67 (Fig. 1a): What represent the red dots? At page 3, line 94: Remove the second "above" word. At page 3, line 96: ECH must be defined before. At page 4, line 135: What kind of sample? Please mention. At page 8, line 225: I suggest to the authors to improve the "Conclusions" section, according with their interesting results.Author Response
Please see the attachment.
